Migratory orientation in a narrow avian hybrid zone

Toews David P.L. 1 3
Delmore Kira E. 1 4
Osmond Matthew M. 1
Taylor Philip D. 2
Irwin Darren E. irwin@zoology.ubc.ca 1
1 Department of Zoology and Biodiversity Research Centre, University of British Columbia , Vancouver , British Columbia , Canada
2 Department of Biology, Acadia University , Wolfville , Nova Scotia , Canada
3 Current Address: Fuller Evolutionary Biology Program, Cornell Lab of Ornithology , Ithaca , NY , United States of America
4 Current Address: Max Planck Institute for Evolutionary Biology , Plön , Germany
Edwards Scott
Electronic publication date: 2017 Apr 18
Publication date: 2017
Volume: 5
Electronic Location ID: e3201
Received 2016 May 1; Accepted 2017 Mar 18
Copyright: ©2017 Toews et al.
Copyright year: 2017
Copyright holder: Toews et al.
License: This is an open access article distributed under the terms of the Creative Commons Attribution License, which permits unrestricted use, distribution, reproduction and adaptation in any medium and for any purpose provided that it is properly attributed. For attribution, the original author(s), title, publication source (PeerJ) and either DOI or URL of the article must be cited.
License URL: https://creativecommons.org/licenses/by/4.0/

Keywords: Migration, Hybrid zone, Migration orientation, Hybrids

Funding: Natural Sciences and Engineering Research Council of Canada 311931 Alberta Conservation Association’s Grants in Biodiversity Research funding was provided by the Natural Sciences and Engineering Research Council of Canada (Discovery Grant 311931 to DEI; CGS-D to DPLT and KED) and the Alberta Conservation Association’s Grants in Biodiversity. The funders had no role in study design, data collection and analysis, decision to publish, or preparation of the manuscript.

==============================
Background

Zones of contact between closely related taxa with divergent migratory routes, termed migratory divides, have been suggested as areas where hybrid offspring may have intermediate and inferior migratory routes, resulting in low fitness of hybrids and thereby promoting speciation. In the Rocky Mountains of Canada there is a narrow hybrid zone between Audubon’s and myrtle warblers that is likely maintained by selection against hybrids. Band recoveries and isotopic studies indicate that this hybrid zone broadly corresponds to the location of a possible migratory divide, with Audubon’s warblers migrating south-southwest and myrtle warblers migrating southeast. We tested a key prediction of the migratory divide hypothesis: that genetic background would be predictive of migratory orientation among warblers in the center of the hybrid zone.

Methods

We recorded fall migratory orientation of wild-caught migrating warblers in the center of the hybrid zone as measured by video-based monitoring of migratory restlessness in circular orientation chambers. We then tested whether there was a relationship between migratory orientation and genetic background, as measured using a set of species-specific diagnostic genetic markers.

Results

We did not detect a significant association between orientation and genetic background. There was large variation among individuals in orientation direction. Mean orientation was towards the NE, surprising for birds on fall migration, but aligned with the mountain valley in which the study took place.

Conclusions

Only one other study has directly analyzed migratory orientation among naturally-produced hybrids in a migratory divide. While the other study showed an association between genetic background and orientation, we did not observe such an association in yellow-rumped warblers. We discuss possible reasons, including the possibility of a lack of a strong migratory divide in this hybrid zone and/or methodological limitations that may have prevented accurate measurements of long-distance migratory orientation.

Introduction

Breeding and wintering ranges of many species are separated by thousands of kilometers. Long-distance seasonal migration exhibited by many taxa moving between these disjunct areas is a complex and energetically demanding task that has been studied for decades (Berthold, 1996). However, the physiological mechanisms, controls and senses involved in navigation of many taxa are still unclear, as are the contributions of various cues to migratory behavior (Berthold & Terrill, 1991; Alerstam, 2006). Much of our understanding of migratory directionality and navigation comes from studies of blackcap warblers (Sylvia atricapilla) in Europe. In an influential series of studies, Helbig and colleagues (Helbig, 1991; Berthold et al., 1992; Helbig, 1994; Helbig, 1996) reared blackcaps from populations that exhibited different migratory routes. They found that, even in captivity in a common environment, populations recapitulated their natural routes, as assayed by Emlen funnels (Emlen, 1970). These studies also found that lab-crossed hybrids between the parental types tended to show intermediate migratory orientations.

Few studies have collected similar data in wild populations, primarily due to logistical challenges. Such observations are particularly pertinent to studies of migratory divides, where populations that differ in migratory directionality come into contact and interbreed (e.g., Helbig, 1991; Bensch, Andersson & Åkesson, 1999; Bensch et al., 2009; Irwin & Irwin, 2005; Ruegg, 2007; Irwin, Irwin & Smith, 2011; Rohwer & Irwin, 2011; Ilieva et al., 2012). For example, if hybrids between divergent populations have a mixture of alleles responsible for migration and exhibit an intermediate migratory orientation, as has been observed in lab-raised blackcap warblers (Helbig, 1991; Helbig, 1994; Helbig, 1996), such a novel phenotype may be inferior and represent an important fitness detriment. This is because, in a number of systems, intermediate routes have been suggested to take hybrid individuals over regions that provide greater challenges in finding food (e.g., deserts) or navigating (e.g., mountains). If so, inferior routes of hybrids can cause lower survival and/or lower fecundity (if arriving on the breeding grounds in poor condition hampers reproduction), thereby contributing to postmating reproductive isolation and promoting further differentiation and speciation of the two populations (Helbig, 1991; Irwin & Irwin, 2005).

To date, the only study to directly assay the migratory behaviour of wild individuals differing in their genetic constitution across a migratory divide is in a study of Swainson’s thrushes (Catharus ustulatus; Delmore, Fox & Irwin, 2012; Delmore & Irwin, 2014; Delmore et al., 2016). In a hybrid zone between coastal and inland forms of Swainson’s thrushes in western Canada, migration was studied using both light-level geolocators, to directly track migration routes and wintering locations (in Central and South America), and orientation chambers to measure migratory direction of birds beginning fall migration. Hybrids exhibited greater variability in their routes than parentals did, with some taking routes that were intermediate to parental forms and took them over the arid regions of the southwestern USA. Moreover, among Swainson’s thrushes within the hybrid zone, genetic background was predictive of migratory route as well as orientation direction, and one particular genomic region showed a particularly strong association with longitude (i.e., west to east) of migratory route and wintering location (Delmore et al., 2016).

Given the prominence of the idea that differences in migratory behavior may cause reproductive isolation within hybrid zones (e.g., Helbig, 1991; Bensch, Andersson & Åkesson, 1999; Bensch et al., 2009; Bearhop et al., 2005; Irwin & Irwin, 2005; Ruegg, 2007; Rolshausen et al., 2009; Rolshausen et al., 2013; Liedvogel, Åkesson & Bensch, 2011; Rohwer & Irwin, 2011; Ilieva et al., 2012), it is important to test whether there is association between genetic background and migratory orientation within a variety of hybrid zones. Here, we provide another direct examination of this question with a large-scale study of birds sampled on fall migration within a hybrid zone between Audubon’s / myrtle warblers (Setaphaga auduboni and S. coronata) in western North America. Within this hybrid zone there are a full range of hybrid genotypes, indicating that multiple generations of hybrids and backcrosses are present and that reproductive isolation is far from complete. However, the narrowness of this hybrid zone and sizeable amounts of linkage disequilibrium in the center of the zone suggest that some form of moderately strong selection maintains it (Brelsford & Irwin, 2009). Assortative mating and other pre-mating reproductive barriers are unlikely to be strong between these taxa (Brelsford & Irwin, 2009; Brelsford, Milá & Irwin, 2011; Toews, Brelsford & Irwin, 2014), implying a potentially sizeable role for post-mating selection against hybrids, possibly based on inferior migratory behaviour. Band recovery and isotopic data from birds outside the hybrid zone suggests the two parental taxa differ in their migratory movements (Fig. 1; Toews, Brelsford & Irwin, 2014; Toews et al., 2014). If migratory traits are genetically controlled and inherited additively–as the studies of blackcaps and Swainson’s thrushes suggest–we expected a correlation of orientation and ancestry in the hybrid zone. Specifically, we predicted that individuals more genetically Audubon’s-like would orient SSW, individuals more myrtle-like would orient SE, and hybrids would orient intermediate between these two (i.e., south).

Figure 1 Distribution and band recoveries of Audubon’s and myrtle warblers, and location of study site.

(A) Banding data obtained from Brewer et al. (2006) and the Canadian Bird Banding Office (2013, unpublished data). Note that there is a distinct subspecies of Audubon’s warbler, the black-fronted warbler, that occurs in Mexico and is not distinguished on this map. (B) Sites of capture for migratory yellow-rumped warblers with (C) the site of the orientation assays. Map data: Google, DigitalGlobe.

We used a video-based orientation cage method, initially developed by Fitzgerald & Taylor (2008). These orientation cages are similar in many respects to traditional Emlen funnels (Emlen, 1970), but in this case the behaviour of birds was scored using video cameras. As compared to the scratch marks quantified with Emlen funnel experiments, this method has a number of benefits, including: (1) scoring of the videos can be automated, ensuring objectivity, (2) a longer period of observation can be obtained for each individual, (3) specific time periods can be analyzed in isolation and (4) any behavioural changes over the course of the trial can be quantified. We designed our study to assay the orientation of individuals in the evening of the day they were captured, which previous research suggests is predictive of later orientation (Ilieva et al., 2012). We then genotyped each individual with species diagnostic genetic markers and tested whether there was a correlation between orientation and genetic background.

Methods

Study site and orientation trials

Between August 15th and September 12th of 2011 we captured migratory yellow-rumped warblers (n = 181) near Kananaskis, Alberta, a site at the center of the Audubon’s/myrtle warbler hybrid zone (Brelsford & Irwin, 2009). Our sampling was concentrated in two areas: along the southern edge of Barrier Lake (51.0235°N, 115.0608°W) and within the hamlet of Lac Des Arc (51.0536°N, 115.1589°W; Fig. 1C). We set up mist-nets before dawn and used passive netting along with song playback (using a tape of a variety of yellow-rumped warblers as well as begging calls from nestlings; previous experience showed that similar recordings could be used to attract all phenotypes in the hybrid zone) to increase our likelihood of catching individuals of our target species. These sites, timing, and capture methodology were chosen to capture birds that were beginning their migratory movements, but it is possible that some of the individuals sampled were close to their breeding territories or natal areas. Immediately after capturing each individual we took morphometric measurements (bill, tarsus, wing and tail length; according to Pyle, 1997), photographs, a blood sample (10–40 µL), and applied a unique aluminum USFWS leg band. Birds were then transported approximately three kilometers from the capture site to the location where the orientation trials were performed (51.0286°N, 115.0242°W). This site is a large, recently clear-cut field (∼400 m2; <10 years old) near the Kananaskis Biogeoscience Institute and has clear views in all directions.

Each individual was placed individually into one of 12 outdoor holding/orientation cages (Fig. 2) and given water and small mealworms throughout the day. The cages were a modified design based on Fitzgerald & Taylor (2008), who used similar cages to study orientation in yellow-rumped warblers (Fig. 2). The cages were leveled, oriented with a compass, and spaced approximately 3–5 m apart. The cage frames were made out of pine boards, with the top and bottom of the cages made from composite plywood. The perch was made with a 9″ plastic embroidery hoop, held up by four 7/16″ dowels, and placed horizontally such that birds could perch on the hoop (Fig. 1C). Since research suggests that birds may use magnetic cues for navigation, the wood pieces were joined with non-magnetic brass screws and stainless steel staples were used to affix screen mesh to the top and sides of the cage (Alerstam, 1993; Muheim, Moore & Phillips, 2006). During the day the sides of the cages were covered with a blue, opaque tarp for sun shelter, with part of the tarp rolled up to provide sufficient airflow (Fig. 2A). Using the same cages to both hold the individuals during the day and run the orientation trials in the evening has a number advantages over traditional use of Emlen funnels (Emlen, 1970). Most importantly, this likely reduces the stress inherent in holding individuals indoors in artificial environments and moving individuals between holding and assay cages shortly before the trials, as is necessary with funnels (Emlen, 1970). We were careful in assessing the welfare of the birds: if individuals did not immediately fly to the perch, showed any signs of stress (e.g., panting), or were not eating properly they were released during the day (less than 15% of birds captured).

Figure 2 Holding and orientation cages as modified from Fitzgerald & Taylor (2008).

(A) Orientation cage with tarp shield. (B) Orientation cage without tarp shield. (C) Dimensions.

Before we started each orientation trial, we removed the food and water dishes and closed each tarp around the sides of each cage, allowing each individual to see only out of the top of the cage, which had a full view of the sky. Through a hole in the bottom of each cage we attached a D-Link Wireless Network Camera (DCS-932L) pointing directly up, with the top of the camera oriented northwards. Unlike cameras that point down (e.g., Muheim et al., 2014), these upward-facing cameras do not obstruct a bird’s view of the sky. The cameras we used have an infrared LED light for illumination during low light conditions. This light was applied consistently across all of the trials, and we are aware of no evidence that birds can see infrared light. The cameras were set such that the video image was recorded with the right side of the video representing the west side of the cage and the left side of the video representing the east side of the cage. We recorded 320p x 240p 30 frame-per-second video from each cage simultaneously using the D-ViewCam software on a PC laptop via a D-Link router. We began recording approximately one hour before sunset and ran the trials until approximately 30 min past sunset. Each evening we ran the trials until the last individual stopped moving, after which we released all of the individuals. We tested each bird once and recorded the behaviour of one to 10 individuals each evening, averaging six birds per evening, with a total of 124 individuals for which we collected complete orientation data. All animal care and experimentation was conducted according to the University of British Columbia protocol Nos. A11-0054 (Project title: Orientation in migratory songbirds) and A09-0131 (Project title: Geographic variation in birds of western Canada). Field permits were provided by the Canadian Wildlife Service, Prairie and Northern Region office (AB Scientific Permit 11-AB-SC023) and Alberta Parks (11–107).

Video analysis

From the D-ViewCam software we exported each video in “.asf” format, noting the start and end time of each trial. We analyzed the video data with radR, an open source platform developed for acquiring and analyzing radar data (Taylor et al., 2010) that was more recently adapted to analyze video files. In brief, radR uses contrast to score individual pixels, and then uses movement, area and intensity to define objects. We sampled the videos at three frames per second. For each detected object we extracted the X and Y coordinates of its centroid, as weighted by the area of the object. Given that the light conditions change over the evening, we used three groups of parameters to accurately and consistently identify the bird as the primary object relative to the background (see Supplemental Information 1 for parameters used). We defined the center of the circular perch as the center of our analytical coordinate system and created an exclusion zone within the circular perch, such that the program would identify the bird only in the region from the perch outwards (Fig. 3). This makes our results more comparable to previous funnel studies, which only recorded jumps where the individual contacts the side of the funnel (Fitzgerald & Taylor, 2008). From this data radR generates a list of time-stamped X and Y coordinates for points where it identified an object (see Table S1 for an example of data extracted for a single individual).

Figure 3 Example of the radR interface.

The image shows two frames from a video taken from the bottom center of the cage. The perch exclusion zone is the area within the circular perch.

Using R 3.0.3 (R Development Core Team, 2013) we applied a number of additional filters, primarily to remove noise (i.e., objects that were not the bird) and remove times when the bird was not moving (i.e., sitting on the perch). To remove noise, we first removed all data where there were three or more objects identified at a single time-stamp, as three or more objects were invariably an artifact of background noise. Sometimes radR identified the bird as two separate objects, primarily when the individual was above one of the four perch dowels, such that parts of the bird stuck out on either side of the dowel. For time-stamps with two objects, we averaged the XY values that were less than 500 pixels apart to include only those times that an individual was above the dowels (i.e., the dowels are slightly less than 500 pixels wide in the video image) and visually inspected the data to confirm this was accurate. To ignore points where the bird was sitting motionless on the perch we removed consecutive time points that had a lower XY distance than a predetermined threshold, which we calculated by studying videos of individuals sitting quietly on the perch (50 pixels for the current analysis). Each of the points that passed these filters were then transformed into an angle relative to North (N = 0°, E = 90°, S = 180° and W = 270°) and given a timestamp (in seconds) relative to the time of sunset for that evening.

From these data we estimated three behavioural traits for each individual: mean orientation, rho and activity, defined as follows. We calculated mean orientation using the “circular statistics” package in R (Agostinelli & Lund, 2013). We used the mean of angles observed over the entire observational period (restricting the time range to only those times when the birds were more active did not meaningfully change the results). We used the same R package to estimate rho, a measure of angular variance that varies between 0 and 1 (i.e., a measure of the concentration of points, with a value of 1 being perfectly concentrated). Finally, we used the total number of time points at which an object (i.e., a moving bird) was detected by radR over the trial as the total activity for each individual.

To assess robustness of estimating the mean orientation for each individual using the data generated from radR, we chose five orientation videos at random and scored them by eye, blind to the output from radR. For this we visually estimated the angle of the bird (if present in the frame) every 30 s, over the entire video, and also recorded whether the bird was on the perch or in mid-flight. We then calculated the difference in angle between this estimate and that obtained from radR for the full dataset and also from only those points where the bird was observed to be in flight. Our data filtering appeared to be effective as our data from a random selection of videos analyzed by eye (for five birds) was consistent with the output from radR. When we compared all of the points observed by eye, including those where the individual was sitting on the perch, the resultant mean angles were within ±22° relative to the output from radR after filtering. If we included only those points where the individual was in flight (by excluding those times we observed by eye for which the individual was on the perch), which our filters within radR were designed to remove, the resultant mean angles were within ±11° of the radR output.

Molecular analysis

Blood samples were taken using a small needle and capillary tube from the brachial vein, stored in Queen’s lysis buffer (Seutin, White & Boag, 1991), and left at ambient temperature until returned to the laboratory. DNA was extracted using a standard phenol-chloroform procedure and resuspended with 50–200 µL of buffer (depending on the size of the pellet) containing 10 mM Tris–HCl and 1 mM EDTA, at pH 8.0, and stored at 4 °C. We sexed individuals molecularly using the procedure described in Fridolfsson & Ellegren (1999). We genotyped individuals at three molecular markers. The full PCR and genotyping protocol for two of the three nuclear markers (CHD1Z and numt-Dco1) was presented in a previous publication (Brelsford & Irwin, 2009). For the third nuclear marker, an 850 base pair fragment of RIOK2, we used the forward primer 5′-ATGGGTGTTGGCAAAGAATC-3′, the reverse primer 5′-GCTCCTCTTCRTTWGCAACA-3′, and a PCR annealing temperature of 60 °C. The enzyme XmnI cuts an allele common in Audubon’s warblers, but leaves intact an allele common in myrtle warblers. To generate a genetic hybrid index we scored a zero for each Audubon’s allele and a one for each myrtle allele for the three markers and divided this by the total number of alleles (six for males, four for females) resulting in an index that ranges between 0 (all Audubon’s alleles) to 1 (all myrtle alleles).

Combining genotype and migratory behaviour

To test whether orientation is associated with genetic background, we used two statistical approaches. First, we used a circular linear model to test whether orientation varied linearly with hybrid index (Agostinelli & Lund, 2013). Second, we used a circular ANOVA to test whether mean orientation angle differed in any way (not necessarily linearly) among these five genetic groups: those with all Audubon’s alleles (h-index = 0; Group A), those with mostly Audubon’s alleles (0 < h-index < 0.5; Group AH), those with mixed genotypes (h-index = 0.5; Group H), those with mostly myrtle alleles (0.5 < h-index < 1; Group MH), and those with all myrtle alleles (h-index = 1; Group M). While the linear model would assume that each hybrid class as an intermediate migratory orientation compared to the two classes on either side of it, the ANOVA does not make that assumption, allowing situations in which hybrid classes might have more extreme orientations than either parental group.

To test whether there was a significant mean orientation of all of individuals considered together, regardless of their genetic background, we used a Rayleigh test. This is a procedure to test the null hypothesis that the orientation angles are distributed randomly, with the alternative being that the distribution is clumped in certain direction(s). The test statistic is r, an estimate of rho, and is the magnitude of the mean vector (Fitzgerald & Taylor, 2008).

Results

Molecular data

We obtained genotypes from 166 of the 181 yellow-rumped warblers captured at the beginning of fall migration through the Kananaskis area, including 123 individuals with orientation data. Based on the three genetic markers, our data set consisted of individuals with a broad range of genetic backgrounds, spanning from Audubon’s warblers (h-index = 0) through a broad array of hybrid genotypes to myrtle warblers (h-index = 1), consistent with the expectation based on prior research of a broad mixture of hybrid classes (F1’s, F2’s, backcrosses, etc.) and pure-type individuals (Brelsford & Irwin, 2009). The daily composite allele frequencies varied relatively little over the study period: the daily proportion of myrtle alleles was usually between 0.3 to 0.6, with an average over the study period of 0.4 (Fig. 4A). Our molecular sexing of individuals indicated that we captured an excess of male birds with 73% of all of the individuals identified as males (Fig. 4B). This was likely due to our use of song playback during mist netting.

Figure 4 Average hybrid index and sex proportion is relatively constant over the migratory period.

(A) Average hybrid index over the fall migratory period. Hybrid index was based on three nuclear genetic markers and was equivalent to the proportion of myrtle alleles (i.e., 0, all Audubon’s alleles; 1, all myrtle alleles). (B) Proportion of daily sample that were male, as determined by molecular sexing.

Orientation trials

In the orientation trials, initial observations of yellow-rumped warblers in the orientation chambers indicated that their activity began to increase approximately one hour before sunset. During this time of increased activity their behavior also changed qualitatively, from primarily sitting on the perch and/or flying occasionally to and from the bottom of the cage, to performing more short flights from the perch to the top of the cage, consistent with zugunruhe (i.e., migratory restlessness; Emlen, 1970). The activity of the birds increased to a peak at around 20–30 min before sunset (Fig. 5), and then gradually declined. Following sunset their activity sharply declined, such that we recorded virtually no movements after 40 min post-sunset. At the end of most evenings individuals usually stopped moving within 5–10 min of each other with remarkable consistency (the steep decline to the right of Fig. 5).

Figure 5 Activity of birds over the evenings during the orientation trials.

Each point is the number of objects identified by radR over a 5-minute time period with the time relative to sunset. The grey points are the raw data from all individuals. The connected, filled circles are the averaged points over a 20-minute window. Peak activity occurs approximately 20 min before sunset.

Of the 123 individuals where we had genotype and orientation data, we found that 96 birds showed strong evidence of orientation behavior (i.e., within-individual r > 0.1; total activity >500). We used all 123 individuals in analyses; analyses using only those individuals with strong orientation did not qualitatively change results. Of the five genetic groups, our sample contained n = 6 for Group A (i.e., genetically Audubon’s), n = 45 for Group AH, n = 41 for Group H, n = 22 for Group MH and n = 9 for Group M (i.e., genetically myrtle). We did not detect a significant difference in mean orientation between the genetic groups (F = 1.23, P = 0.09) and there was no significant linear relationship between mean orientation and genetic hybrid index (t = 0.779, P = 0.22). Hence we could not reject the null hypothesis of no association between genetic background and orientation.

While there was considerable variability in orientation among individuals, there was a significant mean orientation towards 25° or NNE (Fig. 6A; n = 123, among-individual r = 0.320, P < 0.01), with a 95% confidence interval between 5° and 46°. Separating the birds by their genetic hybrid index, each group had mean orientations similar to N or NE, although only Groups M (h-index = 1) and H (h-index = 0.5) showed evidence of significant mean orientations compared to the null of an even distribution of orientation angles: Group A, 38° (n = 6, r = 0.634, P = 0.09); Group AH, 3° (n = 45, r = 0.230, P = 0.09); Group H, 54° (n = 41, r = 0.373, P < 0.01); Group MH, 359° (n = 22, r = 0.321, P = 0.10); and Group M, 15° (n = 9, r = 0.733, P < 0.01).

Figure 6 Orientation of yellow-rumped warblers in relation to genetic ancestry.

(A) Orientation of all individuals in the study. (B–F) Orientation of individuals grouped according to genetic hybrid index.

Discussion

Here we have provided the second study of potential association of genetic ancestry and migratory orientation in an avian hybrid zone. Our use of video-based orientation trials and automated video analysis enabled us to sample a large number of individuals during fall migration while also gathering high resolution and orientation chamber movement data for each bird, a benefit over previous Emlen funnel methods. Despite these methodological advantages and prior evidence for a possible migratory divide across the Audubon’s / myrtle warbler hybrid zone (Toews, Brelsford & Irwin, 2014), we did not find an association between genetic background and orientation. We discuss below the possibilities that this result was due to (1) lack of any true association between genetic background and migratory route within this hybrid zone, (2) short-term orientation during the time of the study not being indicative of long-distance migration orientation, or (3) methodological limitations that hindered the detection of a true association.

First, the prior evidence for a migratory divide across the yellow-rumped warbler hybrid zone is mixed, raising the possibility that our results are due to no actual association of genetic ancestry and migratory differences in the center of the zone. Banding data and wintering range maps of Audubon’s and myrtle warblers indicate that Audubon’s warblers largely winter in the southwestern USA, whereas myrtle warblers mostly winter in the southeastern USA (Toews, Brelsford & Irwin, 2014). However, sizeable numbers of myrtle warblers winter in the southwest (Hunt & Flaspohler, 1998). It has been postulated that most of these belong to a distinct form of myrtle warbler (the subspecies hooveri) breeding in Alaska, the Yukon, and northern British Columbia (McGregor, 1899; Toews, 2017). However, it is also possible that some myrtle warblers from Alberta also migrate to the southwestern USA, and that the current hybrid zone between Audubon’s and myrtle warblers in western Alberta does not correspond to a migratory divide. Isotopic data (Toews, Brelsford & Irwin, 2014) is mostly supportive of the hybrid zone coinciding with a migratory divide, because sites just outside of the hybrid zone differed in their hydrogen isotopic signatures in a way consistent with the expected difference between southwestern and southeastern USA, and birds within the hybrid zone showed a broader mixture of signatures. Taking this evidence together, it is likely that there is at least some sort of transition in average migratory orientation between Audubon warblers in central British Columbia and myrtle warblers in central Alberta, but it is possible that that the transition zone is very broad, with birds in the center of the hybrid zone having no or only a weakly detectable relationship between ancestry and migratory orientation.

Second, it is possible that the behavior that we observed in the chambers is indicative only of short-term and short-distance orientation rather than longer-term and longer-distance migratory orientation. Among individuals we found a lot of scatter in directional tendencies, with a mean migratory orientation that was surprisingly towards the NNE (26°), odd for birds on fall migration, which is expected to be generally southward. A possible explanation is that individuals may be moving northeast, out of the valleys in the Rocky Mountains, to later turn south. The orientation of the valley near the capture location and orientation experiment area is approximately 23° (NNE; estimated using Google Earth™), very close to the observed orientation of the birds when grouped together (26° NNE). Our sample likely included many hatch-year birds, raising the possibility that the northward orientation of some birds may be indicative of regional-scale post-fledging movements (such as those made by blackpoll warblers; Brown & Taylor, 2015), the incidence and function of which are still unknown. While previous orientation studies have tested the effect of ecological barriers on migratory behaviour (i.e., water bodies; Sandberg & Moore, 1996; Ilieva et al., 2012), this is one of the first studies to assay orientation in and around mountainous areas. Using high-resolution radar technology, Williams et al. (2001) found evidence that nocturnal migrants responded to local topological features by changing their orientation during fall migration, especially those birds migrating below 300m, as is assumed with yellow-rumped warblers. Given the type of data these methods collect, however, it is challenging to assign these types of observations to specific species or even species groups (Williams et al., 2001). For the warblers in our study, individuals might have a memory of the axis of the valley at the time of capture, and then orient in that direction. We recommend that future orientation studies around mountainous areas should consider including additional orientation localities in valleys of varying orientations. It would also be useful to assay individuals each evening over a longer period (i.e., 1–2 weeks) to test whether this orientation is maintained or dissipates with time. This could provide a robust test of the role of topological features in influencing migratory movements.

Finally, it is possible that the observed chamber orientations are not representative of movements the birds would make if they were outside of the chambers. All experiments using orientation chambers with captive birds run the risk that migratory movements in specific systems may not be well represented by cage movements, although it is remarkable how often there is a strong association (e.g., Helbig, 1991; Helbig, 1996; Alerstam, 1993; Van Doren, Liedvogel & Helm, 2017). The movements and pattern of activity of the warblers in our study were qualitatively similar to previous descriptions of zugunruhe, suggesting that the yellow-rumps in our sample were expressing behaviours consistent with migration. But given the unexpected mean orientation toward the northeast, the lack of any relationship with genetic background, and the reasonably short duration of the movement behavior each evening, we acknowledge that it is plausible that orientation cage behaviors in our study are not indicative of orientation in free-moving birds. Given this concern, we suggest that tracking studies using direct tracking technology such as radio towers and miniaturized geolocator tags may be more useful for studying large-scale migratory orientation (Taylor et al., 2011; Delmore, Fox & Irwin, 2012; Veen, 2013; Delmore & Irwin, 2014).

In conclusion, we used molecular genetic methods and a video-based orientation assay that provided objective, high resolution temporal data for many individuals in a semi-natural setting. Our results show no significant association between genetic ancestry and chamber orientation of migrating yellow-rumped warblers within a hybrid zone that was postulated to correspond to a migratory divide. These results provide an interesting contrast to Swainson’s thrushes, in which genetic ancestry within a hybrid zone is predictive of migratory orientation as measured both by geolocators and orientation chambers (Delmore et al., 2016). We suggest that, with future modifications, these methods could provide a powerful tool for understanding migratory orientation in many species. In particular, when replicated in multiple locations, one could use these orientation assays to examine the effect of local topographical features. Or, if assayed orientations were verified by geolocators, one could identify potentially maladaptive migratory tendencies in hybrid individuals. We hope these methods will contribute to the understanding of migratory divides, and of migration more generally.

Supplemental Information

Supplemental Information 1 Supplementary Information

Click here for additional data file.

Table S1 Supplementary Table

Raw genotype and orientation data.

Click here for additional data file.

We thank Stephanie Cavaghan with assistance in the field. The Kananaskis Biogeoscience Institute graciously provided accommodation and logistical support. We thank John Brzustowski for assistance with implementation of the radR analysis. Permits and land access was provided by the Canadian Wildlife Service, Alberta Fish and Wildlife Service, Alberta Parks and Recreation and the Alberta Sustainable Resource Development Land and Forest Division.

Additional Information and Declarations

Competing Interests

Author Contributions

Animal Ethics

Field Study Permissions

Data Availability

The authors declare there are no competing interests.

David P.L. Toews conceived and designed the experiments, performed the experiments, analyzed the data, wrote the paper, prepared figures and/or tables.

Kira E. Delmore conceived and designed the experiments, performed the experiments, analyzed the data, wrote the paper.

Matthew M. Osmond analyzed the data, wrote the paper, prepared figures and/or tables.

Philip D. Taylor contributed reagents/materials/analysis tools, wrote the paper.

Darren E. Irwin conceived and designed the experiments, contributed reagents/materials/analysis tools, wrote the paper.

The following information was supplied relating to ethical approvals (i.e., approving body and any reference numbers):

UBC Animal Care Committee Nos. A11-0054 (Project title: Orientation in migratory songbirds) and A09-0131 (Project title: Geographic variation in birds of western Canada).

The following information was supplied relating to field study approvals (i.e., approving body and any reference numbers):

Canadian Wildlife Service, Prairie and Northern Region Scientific Permit 11-AB-SC023 and Alberta Parks 11-107.

The following information was supplied regarding data availability:

The raw data has been supplied as a Supplemental Dataset.

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
