# Peer review of "Migratory orientation in a narrow avian hybrid zone"

_PeerJ, doi:10.7717/peerj.3201_

## Round 0.1 · original submission · Major Revisions

The two reviews I have received are both very thorough. Whereas reviewer 1 sees some positives in the paper, reviewer 2 is very concerned that the observed data represents artifacts. In your revision, you need to address the possibility that the data collected represents artifacts and not real data, and rebut this as well as you can. Otherwise the paper should surely be rejected.

Please also address reviewer 2's comment about selection against hybrids. Reviewer 1 also had comments on this point. Your revision will definitely be sent out for re-review.

Reviewer 1 ·

Basic reporting

The study should provide more information on the frequency of hybrids known for this population. If hybridization is very common and hybrids and backcrosses are more common than pure genotypes, then the prediction that hybrids may show maladaptive behavior might need to be revised. I elaborated on this idea in my comments for the authors.

Experimental design

The study is conducted in a single site, which is a strength but also a limitation in this kind of research. To some this would mean a fatal lack of replication. I believe the authors can discuss this problem but it needs to be clearly pointed out in the paper.

Validity of the findings

Related to my comment above on experimental design, the paper reports a negative result, but it was done at a single site. This may represent lack of replication and requires appropriate discussion in the paper. I believe it is not a fatal problem if the authors clearly recognize the limitations of the study. I elaborated on this issue in my letter to the authors.

Additional comments

The most evident strength of the paper is the description of a refined method for scoring orientation in the field, which solves many problems of the many former approaches based on Emlen funnels. Otherwise the study reports negative results, which are interesting but need to be properly framed and described to avoid confusion with conclusions being drawn on true lack of effects. My most important criticisms go precisely to this central issue. In order to tackle with it properly, the paper needs at least the following major modifications:

1. The title should be changed to clearly state that the result (lack of the expected effect of genotype on orientation) may be just a local occurrence, rather than a general phenomenon across the hybrid zone. Most evidently, the birds could orient differently according to their genetic background in another place on the contact zone that is less influenced by the mountains. From this perspective, the study lacks replication and the negative result turns out to be weak evidence of true lack of effects. In relation to this, the introduction (and also the discussion) should be revised to clearly state what can and what cannot be tested with this species and setup. The paper needs to make an effort to clearly distinguish true lack of patterns (such as “hybrids between these two genotypes do not show maladaptive migration, as they orient the same as pure genotypes”) from negative results (“possible maladaptive migration of hybrids could not be detected”). Note that in this study negative results could be due to various things, most importantly the possibly “wrong” choice of a single location with strong influence of mountains.

2. The paper needs to test for age effects. Birds were aged in the field but no use is apparently made of age data. However, first year birds probably show innate orientation while older birds enrich their behavior with previous migration experience (which is admitted by the authors; L. 343-344). This issue requires statistical treatment or at least an explanation as to why age was not included in the analysis despite its recognized importance in studies of the genetic control of migration.

3. The fact that hybrids are more abundant in the sample than pure A and H genotypes somewhat tells that hybrids are not in real disadvantage. This should be discussed appropriately. If this circumstance was already known before the study, it should be put forward already in the introduction. It does not matter if this information would change the predictions of the study (maladaptive migration of hybrids would be less expected in a contact zone where mixed genotypes make 88% of captured birds). This information is important to appropriately frame the study of migration in relation to genotype under the state-of-the-art paradigm of hybrid disadvantage.

Specific comments:

L62-63: “our understanding of how these migratory behaviors are expressed under natural settings is unclear.” This is true, and this paper is not going to contribute much on this regard given that cages are not natural settings. It is true the birds are expressing their behavior in cages at the place where expressing this behavior makes sense, but for some questions more controlled settings may be more advisable. I wonder if Helbig’s blackcaps would have showed the same behavior in a setting of the kind used in this paper.

L131. Song playback was used to attract birds into mist-nets. Pease clarify which song was used, if it belonged to Audubon’s or myrtle warblers. Hybrids that respond to one but not the other parental song (if that is possible) could also have the parental migration pattern... or not, in any case the information is relevant to correctly evaluate the method. The issue deserves a comment, given the fact that tape luring usually creates various kinds of bias in this type of sampling (such as in relation to sex in this case, but there might be other more cryptic ones).

L. 134. Please specify which age classes were distinguished. I assume these were first year and older birds.

L. 151. Remove one “traditional” (typo).

L. 265-269. I guess these groupings where meant to capture the different degrees of genetic introgression between species that birds may score at the contact zone (hybrids and backcrosses to simplify). Please clarify. If I am right, then the backcrosses are expected to have parental behaviors as they are likely selected through the survival of their parents with “non-lethal” migratory behaviors.

L. 275-277. This is informative but does not say much about the proportion of myrtle alleles of each bird if no information is given on the variance of this average estimate. I would rather produce a graph in which each bird captured is represented by a dot in Fig. 4A, so that the genetic background of each bird (and the variation around this 0.4 estimate) can be clearly visualized.

L. 358-360. Compared to Ilieva’s, this study has the advantage that different genotypes are sampled at the same place, but it has the other problems I explained above: age effects not tested, single site may be good but also leads to lack of replication, etc. Apart from this, in this system hybrids (as identified by their mixed genotype) are very abundant, which implies that hybridization has low impact on fitness via expression of maladaptive migration, either because migration is not genetically controlled in this species or because hybrid behavior is not so disadvantageous compared to the behavior of the parents as it has been reported is studies of other species (especially those this paper is being compared with).

L. 363. The statement “Our data were consistent with selection against hybrids in the non-breeding season” is contradictory with L. 353 “Mean orientation direction did not differ between the different genotype classes”. This contradiction is also seen in the summary (L. 40-41).

Reviewer 2 ·

Basic reporting

The manuscript is structured and written in a very proper way. In this respect in my opinion it does fully meet your requirements.

Experimental design

please see below. No other comments here.

Validity of the findings

I had the opportunity to review an earlier version of this manuscript. At that time I had rather fundamental concerns about the validity of the results. This new version has been improved in many respects and some aspects are explained and discussed in a more appropriate way, however, the main problems I see could not be healed:

(1) Northward orientation during autumn migration: besides all the explanations through reverse migration and behavior at ecological barriers the most probable thing is that the measurements simply are artefacts which seems not to be uncommon in tests of wild birds in Emlen-like setups, no matter whether we look at films, scratches or bill marks for the analysis. The time pattern of the activities with the sharp end in all individuals but one is another indication that any unnatural trigger influenced the birds activities and thus presumably also orientation in the cage. Migration behavior patterns of at least the European warblers look completely different e.g. in radar studies. There is a lot more individual variation in onset and end of the migration activities. I am still convinced: whatever has been measured in this study is rather not a proxy for the natural behavior.

(2) Similar directional preferences: Even if the preferred mean direction should be resulting from a natural trigger (like valley orientation) and not just be an artefact the directional preference of both “pure” species groups are identical and not different from hybrids. Therefore the substrate on which selection could act is the same for all groups and cannot act differently on hybrids and “pure” species individuals. Thus I do not understand how this is “consistent with selection against hybrids on migration” (as said e.g. in the abstract).

---

## Round 0.2 · accepted · Accept

Thank you for the revised manuscript. I feel you have done your best to answer the reviewers' comments and have made a genuine effort to present your results in an unbiased fashion, whether significant or not. For this reason I have elected not to send it out for re-review and to simply recommend acceptance.